# Advancing Medical Assistance: Developing an Effective Hungarian-Language Medical Chatbot with Artificial Intelligence

Barbara Simon [1], Ádám Hartveg [1], Lehel Dénes-Fazakas [1,2,3], György Eigner [1,2,*] and László Szilágyi [1,2,4]

1 Physiological Controls Research Center, University Research and Innovation Center, Obuda University, 1034 Budapest, Hungary; simon.barbara@uni-obuda.hu (B.S.); hartveg.adam@uni-obuda.hu (Á.H.); denes-fazakas.lehel@uni-obuda.hu (L.D.-F.); szilagyi.laszlo@uni-obuda.hu (L.S.)

2 Biomatics and Applied Artificial Intelligence Institute, John von Neumann Faculty of Informatics, Obuda University, 1034 Budapest, Hungary

3 Doctoral School of Applied Informatics and Applied Mathematics, Obuda University, 1034 Budapest, Hungary

4 Computational Intelligence Research Group, Sapientia Hungarian University of Transylvania, 540485 Tîrgu Mureș, Romania

* Correspondence: eigner.gyorgy@uni-obuda.hu

**Abstract:** In recent times, the prevalence of chatbot technology has notably increased, particularly in the realm of medical assistants. However, there is a noticeable absence of medical chatbots that cater to the Hungarian language. Consequently, Hungarian-speaking people currently lack access to an automated system capable of providing assistance with their health-related inquiries or issues. Our research aims to establish a competent medical chatbot assistant that is accessible through both a website and a mobile app. It is crucial to highlight that the project's objective extends beyond mere linguistic localization; our goal is to develop an official and effectively functioning Hungarian chatbot. The assistant's task is to answer medical questions, provide health advice, and inform users about health problems and treatments. The chatbot should be able to recognize and interpret user-provided text input and offer accurate and relevant responses using specific algorithms. In our work, we put a lot of emphasis on having steady input so that it can detect all the diseases that the patient is dealing with. Our database consisted of sentences and phrases that a user would type into a chatbot. We assigned health problems to these and then assigned the categories to the corresponding cure. Within the research, we developed a website and mobile app, so that users can easily use the assistant. The app plays a particularly important role for users because it allows them to use the assistant anytime and anywhere, taking advantage of the portability of mobile devices. At the current stage of our research, the precision and validation accuracy of the system is greater than 90%, according to the selected test methods.

**Keywords:** medical assistant; chatbot; health; Hungarian

## 1. Introduction

### 1.1. Medical Chatbot Assistants

Medical chatbot assistants are a new way to help the field of medicine and its development [1–5]. Usually, they can be accessed and used via a website or through a mobile application. These solutions use artificial intelligence (AI) in the background to provide the right answers to user questions. Nowadays, many branches of information technology are involved in healthcare, so it is not surprising that the use of medical chatbot assistants has started to spread. Numerous medical chatbots have been developed, each with their own advantages. Despite their widespread adoption elsewhere, the utilization of such tools is not yet prevalent in Hungary. Furthermore, the absence of a medical chatbot capable of communicating in Hungarian underscores the need for our initiative. Our future aim

is to create an official and seamlessly functioning medical chatbot that facilitates continuous communication. In this article, we present our first results of this research objective by introducing the AI-related solutions developed when a small text corpus is available. From the perspective of the operation of solutions, users who enter their symptoms should receive prompt advice and answers through this platform. The use of chatbots can have a number of positive impacts on healthcare [6]. One of the most important benefits is that they reduce the workload of healthcare professionals [7–10]. This will help the overall efficiency of healthcare facilities in the long run. Another very important advantage is that a chatbot can provide an immediate answer to patient questions and concerns, so that they can get help at any time of the day.

### 1.2. Natural Language Processing

Natural language processing (NLP) [11–15] is a subset of AI solutions dedicated to exploring the interaction between human language and computers. It employs various methods to comprehend and interpret text, with the aim of generating written content indistinguishable from human-authored pieces. The breadth of tasks in NLP includes tokenization and named entity recognition, which involve breaking down text into smaller units, such as words or sentences, known as tokens, facilitating further analysis and processing [16–20]. Assigning grammatical tags to words aids in understanding their roles in a sentence, enabling the identification and interpretation of their grammatical properties. NLP employs a blend of rule-based approaches, statistical models, and machine learning algorithms. In essence, NLP is the automated analysis and representation of human language for computers, using theoretically grounded computational techniques [21].

### 1.3. Challenges in Developing Hungarian-Language AI Tools

The development of AI tools for the Hungarian language presents unique challenges that extend beyond the typical complexities encountered in more commonly supported languages. The primary difficulties are twofold: linguistic complexity and resource scarcity.

Hungarian is an agglutinative language with a complex morphological structure, making it challenging for natural language processing (NLP) [22]. It requires sophisticated algorithms to parse sentences accurately due to its rich inflection and morpheme complexity. Additionally, Hungarian suffers from a lack of large, annotated datasets necessary for training AI models, which hampers the development of effective tools for tasks like speech recognition and machine translation. This resource scarcity results in less reliable and lower-performing AI applications for the language [23].

In response to these challenges, our research has been focused on creating robust AI tools specifically designed for the Hungarian language. One of the major contributions of this study is the development of a specialized morphological analyzer that tackles the complex inflection characteristic of Hungarian. We have also constructed a comprehensive dataset from the ground up. This dataset is annotated to assist in training our models, enabling them to process Hungarian with higher accuracy and efficiency [24].

The paper is organized to guide the reader through the entire research process and its implications methodically. Following this introduction, Section 2 reviews related work, shedding light on existing efforts and situating our contributions within the broader academic landscape. Section 3 delves into the methodology, detailing the steps taken in data collection and model training. Section 4 presents our results, offering a critical analysis of the effectiveness of our solutions. Section 5 discusses potential avenues for future research and the expected impact of our work on the development of Hungarian-language AI tools. The paper concludes with a summary of our findings and reflections on the future potential of AI applications in processing agglutinative languages like Hungarian.

## 2. Related Works

### 2.1. Medical Chatbots

Ada Health is a healthcare technology company that uses AI to help people monitor their health, but also to help them make lifestyle changes [25]. Ada Health can be downloaded to mobile phones, where it will diagnose you if you write down your complaints. It can also remember previous diseases and predispositions, which is a very important feature because it gives you more accurate conclusions. In the app, you can ask questions, register symptoms, and get information about your health status. Ada Health is able to ask users detailed questions in order to give a more accurate diagnosis. Ada Health also looks at test results, patient history, and general health to help people understand the possible causes of their illnesses. This application works with a large number of health institutions. This gives them access to official medical information and advice. A study involving 378 patients compared the safety of Ada's emergency advisory system with the Manchester Triage System (MTS) in hospital emergency care [26]. Ada showed a high safety rate in all medical specialties in the emergency department (94.7%), with a particular focus on internal medicine, orthopedics and traumatology, and neurology. Over 43% of patients in the lowest three categories of MTS could have sought less urgent care safely, such as visiting their general practitioner or treating their symptoms at home. With Ada, the workload in emergency departments can be reduced by directing patients who need care to less urgent care at home.

A similar study by Lee and Kang [27] addressed this topic during the COVID-19 epidemic. Given that patients were reluctant to leave their homes and avoided contact by default, they wanted to create a medical chatbot to help patients without having to leave their homes. In terms of data collection, the study used a web-based healthcare platform, the HiDoc, which allowed users to anonymously describe their symptoms. For the dataset, the titles of the posts were collected and presented in a one-sentence format. Data cleaning included eliminating duplicate and missing data, excluding ambiguous sentences, and correcting mislabeled cases, demonstrating a rigorous approach to ensure data quality. Improvements in telemedicine and the proliferation of digital platforms have been accompanied by a reduction in face-to-face interactions between patients and healthcare providers, which became particularly important during the COVID-19 pandemic [27].

Diabot, presented in [28], is a generic and diabetes-specific version of a chatbot that uses NLP techniques based on health data. Diabot interacts with patients and generates specific predictions using the Pima Indian diabetes dataset. The study gives importance to ensemble learning, which combines weak models to create a balanced and accurate model. The ensemble model shows good accuracy in predicting both general health and diabetes. Diabot successfully interacts with all patients and the methods used are incentives for further investigation of ensemble learning. The paper highlights Diabot's simple user interface provided by React UI and compares in detail the performance of different machine learning algorithms [28].

Another relevant example to our research is the so-called Medical ChatBot presented in [1]. The authors have specifically used support vector machine (SVM) technology in their research and compared it with different methods. They chose SVM because of its ability to detect more complex relationships than other classification models. They included various data sources in the training and testing processes, using a 60–40% split between training and testing data [1].

Table 1 provides a succinct comparison of these different chatbots based on key performance and operational metrics, illustrating their effectiveness and user experience. As can be observed, the AI that performs best is the one that can be used on a smartphone. However, Ada also achieves better performance for specific diseases, but its overall accuracy is lower compared to other chatbots.

**Table 1.** Comparison of medical chatbots in recent literature.

| Reference | Functionality | Accuracy | Interface | Integration Support |
|-----------|---------------|----------|-----------|---------------------|
| [25] | Diagnostic support | 0.7 | User-friendly | Healthcare data |
| [28] | Predictive diagnostics | 0.86 | User-friendly | NLU |
| [27] | Specialty matching | 0.96 | Smartphones | Healthcare data system |
| [1] | Query processing | 0.95 | User-friendly | API integration |

*2.2. Ethical Implications*

In recent developments within the realm of AI chatbots, significant attention has been directed towards understanding their ethical implications, particularly in sectors such as education and research. A study by Kooli [29] provides a comprehensive analysis of the challenges and ethical considerations inherent in the deployment of chatbots. This paper highlights issues such as data privacy, informed consent, and the potential for bias, offering solutions to mitigate these risks. Such contemporary analyses not only add to our understanding of the ethical landscape surrounding AI technologies but also underscores the critical need for frameworks that ensure responsible AI usage. This perspective is particularly pertinent to our research as it aligns with our investigation into the implications of AI-driven communication tools in medical settings, where ethical considerations are paramount.

Another study on AI ethics in healthcare [30] offers an in-depth analysis of the ethical and regulatory challenges associated with deploying artificial intelligence (AI) technologies in healthcare settings. It focuses on how AI technologies intersect with privacy and data protection issues, particularly under the stringent regulations of the European General Data Protection Regulation (GDPR). The review highlights the critical importance of compliance with GDPR for AI applications in healthcare, detailing the implications for patient data privacy, consent, and security. The paper also discusses the broader ethical considerations, such as bias, transparency, and the accountability of AI systems in clinical settings. Through its comprehensive analysis, the article aims to inform developers about the essential guidelines and practices for integrating AI into healthcare responsibly, ensuring that these innovations benefit patients while safeguarding their personal information and rights.

**3. Data Collection**

Collecting and organizing health data is always a challenge, especially in the field of medical technologies where data quality is a crucial aspect. Today, health information is critical for the prevention and treatment of diseases. The data for this investigation underwent meticulous curation to ensure quality, originating from databases housing various categorized complaints (e.g., [31–33]), presented in JSON format [34]. Further chatbot JSON dataset examples are indicated in [35,36]. The initial dataset was processed and translated from English to Hungarian. Subsequently, we extracted relevant information from healthcare databases to complement our dataset. From this comprehensive dataset, we conducted further preprocessing steps to create our own dataset tailored to the specific needs of our research [37,38]. We systematically curated our dataset with a rigorous emphasis on diversity, encompassing a broad spectrum of areas, including many prevalent diseases commonly encountered in everyday life. This meticulous approach involved gathering information to establish a well-rounded and inclusive foundation. By incorporating a comprehensive array of diseases that are commonplace in daily experiences, we aimed to enhance the robustness and applicability of our dataset. This strategy strengthened its capacity for meaningful insights and analysis across a wide range of health-related scenarios. We amassed a comprehensive dataset comprising 36 diseases, with uniform data amounts across all classes. This approach ensured parity in the quantity of information available for each disease category, thereby facilitating an equitable assessment of the chatbot's performance in disease detection. The uniformity in data distribution among classes was designed to mitigate potential biases and enhance the model's ability to generalize

effectively across the diverse spectrum of diseases under consideration [39,40]. First of all, we collected textual data that a user would give to a chatbot, i.e., data about symptoms and complaints. Then, we assigned them to the disease they belong to. The final step was to compile the correct cures and treatments for the diseases in a dictionary and collect what the chatbot could answer in these scenarios. In the future, our aim is to expand this database so that as many diseases as possible can be identified and resolved. We would like to collect more sentences for existing diseases and add more diseases to the current database. Health data collection and analysis are always a dynamic process and such projects require a long-term commitment.

As illustrated in Figure 1, the distribution of sentences in our dataset is categorized by disease. This visualization aids in comprehending the breadth and depth of our textual data, which spans across various medical conditions, providing a solid foundation for the AI to learn from real-world examples. The figure shows the extensive coverage of diseases, highlighting the comprehensive nature of our dataset. Figure 1 shows the collected text database where the horizontal axis shows the 36 diseases collected and the vertical axis shows the medically accurate sentences collected, describing the diseases in context.

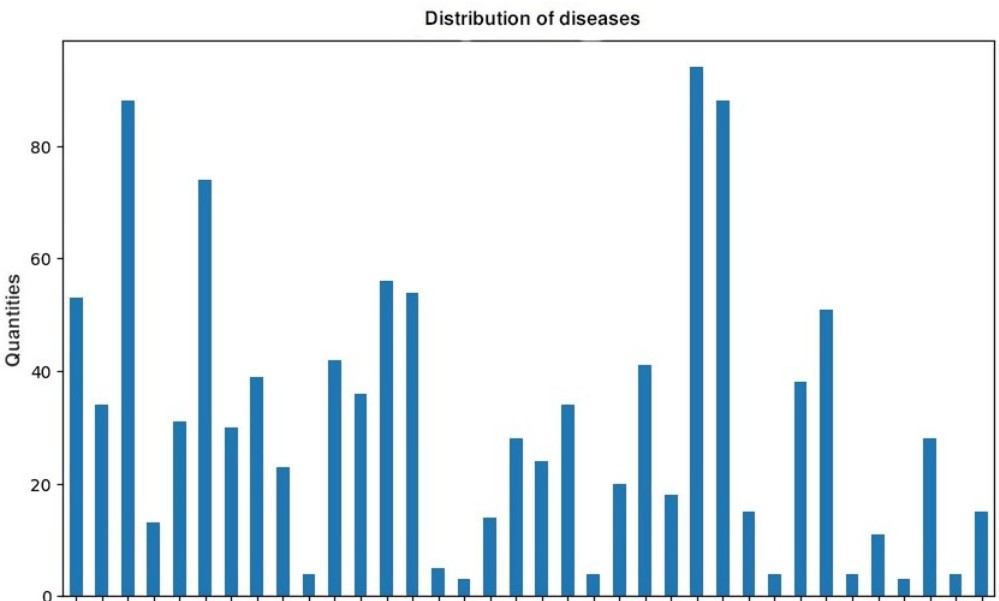

**Figure 1.** Sentence distributions categorised by disease.

## 4. Dataset

The dataset presented in the following is a unique and important resource in the field of disease diagnostics, which we have collected ourselves as part of a project that is still in progress. The data include symptoms related to 36 different diseases, with each complaint being associated with a disease. The data are available in a structured format, in a CSV file, where the first column contains the complaints and the second column contains the associated diseases. The dataset contains a total of 1500 records describing in detail the symptoms of each disease. The data collection process is a meticulous and time-consuming task that involves thorough examination and verification of each complaint–disease association. As our project is still in progress, we have dedicated our efforts to ensuring the accuracy and reliability of the existing 1500 records. This commitment to precision and thoroughness in data collection naturally limits the speed at which we can expand our dataset. We are actively working towards acquiring additional data to enhance the dataset. Furthermore, as our research project is still in progress, we are continuously gathering additional data to enhance the richness and diversity of the dataset. We anticipate that future iterations will include a more extensive range of symptoms and diseases.

Table 2 presents a detailed view of the symptom–disease associations utilized in our research. This table is fundamental for understanding how symptoms are directly linked to specific diseases, which supports the development of our AI model's diagnostic accuracy. For instance, the association of 'inflamed eyes' with 'conjunctivitis' provides a direct insight into the practical application of our dataset in medical diagnostics.

**Table 2.** Some examples from the symptom–disease association table.

| Symptom | Disease |
|---|---|
| My eyes are inflamed. | Conjunctivitis |
| I feel tired and irritable during the day. | Insomnia |
| Warm, red skin over the affected joint. | Arthritis |
| Throbbing in the neck or ears. | High blood pressure |

The dataset quality, particularly the issues of balance and semantic clarity, is a crucial factor and it is important to consider how it impacts machine learning (ML) and deep learning (DL) models. Imbalanced datasets can skew model training, leading to biased outputs and poor generalization to real-world scenarios. Similarly, semantically sloppy datasets, where the data are noisy or inconsistently labeled, can confuse models and degrade their performance. The reference paper, "An alternative approach to dimension reduction for Pareto distributed data: a case study" [41], offers insights that could be relevant to addressing these challenges in the context of Hungarian language processing. Although the paper primarily focuses on dimension reduction for Pareto-distributed data, its methodologies and findings could be adapted to improve the handling of unbalanced and semantically inconsistent datasets in NLP tasks.

Specifically, the authors' approach to dimension reduction, which prioritizes preserving significant variance in highly skewed distributions, could inform techniques for managing datasets where certain linguistic features or labels are disproportionately represented. By integrating such dimensionality reduction techniques, researchers might better manage and interpret large, complex datasets, leading to more robust AI models for languages like Hungarian, whih face data scarcity and quality issues.

## 5. Methods

### 5.1. Long Short-Term Memory

Long Short-Term Memory (LSTM) [42–44] is a specialized deep learning technique designed for analyzing sequential data, addressing issues found in conventional recurrent neural networks (RNNs) [45–47] and other machine learning algorithms. It was proposed by Hochreiter and Schmidhuber [48] to overcome the gradient vanishing problem and enhance the effectiveness of RNNs [49–53]. LSTM enables the retention and utilization of long-term information in a network. There exist four primary elements in this context: the input gate, the forget gate, the introduction of new information, and the output gate. These components play a crucial role in transferring information from one point to another and in retaining and storing past information. The forget gate assesses the degree to which preceding information should be disregarded in the cell state. Meanwhile, the input gate determines how much the cell state should be refreshed with the latest information. The new information outlines the extent to which the cell state should be updated with the current input. Lastly, the output gate dictates the degree to which the cell state should be utilized in generating the output layer.

LSTM is a special version of recurrent neural networks (RNNs) and can detect long-term relationships in text. Therefore, LSTM models are the ideal choice when a chatbot needs to process text data and understand it. For chatbots, the incoming text messages are often sequential and LSTMs can help them to easily process and respond to them. LSTM models can be easily fine-tuned and customized to the specific application. This allowed our chatbot to perform 36 different classification tasks, in our case, for diseases. The more data they are provided with, the better answers they can generate. Furthermore, LSTM models

can be used to describe the grammar and complexity of linguistic and semantic language. Furthermore, these models can preserve and handle long-term textual contexts. This enables chatbots to better understand and interpret the requests and responses provided by their users [54–56].

In developing the medical assistant chatbot, our research team preferred to use LSTM models. Although BERT (Bidirectional Encoder Representations from Transformers) models are highly efficient in natural language processing, our choice of LSTM is justified by a number of factors. First, LSTM models can handle smaller datasets more efficiently. In the present case, since the amount of available medical data was limited, the ease of adaptability and less rigorous demands on the pre-learned data offered by LSTM were necessary. Second, medical texts often contain long-term dependencies that can be key to making correct diagnoses and treatment plans. LSTM models can efficiently handle and memorize these long-term relationships, providing an advantage over BERT [57–59] models. A third reason for choosing LSTM models is the ease of implementation and fine-tuning. While fine-tuning a BERT model often requires complex procedures and significant resources, LSTM models are more flexible and can be more easily fine-tuned on smaller datasets with minimal prior expert knowledge.

### 5.1.1. Composition

First, we created a tokenizer object. A tokenizer is a tool that helps tokenize words. This is important because machine learning models need to use numbers as input and convert words into numbers. First, we called the fit_on_texts() method on our input data. This method initializes the dictionary, which is an empty dictionary of words and their corresponding numbers pairs. The method counts the number of occurrences of each word in the processed text. This helps to determine the importance of the words in the subsequent processing. After the tokenizer object was trained, the texts_to_sequences() method was used to tokenize the input texts. Machine learning models generally require input of the same length. We used the pad_sequences() method to tokenize sequences with the same length, adding zeros to shorter sequences. This way, all input sequences contained the same number of elements. Finally, vocab_size was determined using the trained tokenizer. The tokenizer.word_index contains a dictionary, where the numbers assigned to the words are stored.

### 5.1.2. Layers

The very first layer, as shown in Figure 2, is an embedding layer [60–62], which is a layer in neural networks that transforms the input data into a form that is easy to manage and can be efficiently handled in the network. In text processing, for a language model, it transforms words into vectors that represent those words in a field. In this layer, we first had to define the input dimension. In our case, this input dimension was a value also called "vocabulary size". The vocabulary size is equal to the total number of different individual words in a given dataset. The next parameter that had to be defined was output_dim. This gives the dimension of the output vectors. In our case, it was 100, so the output vector was a 100-dimensional vector representing the input words. The output dim setting played an important role in the embedding layer performance and model efficiency. In general, output vectors with higher dimensions contain more information, but the model becomes more complex and requires more computational resources.

The next layer is an LSTM, where the number of hidden neurons was set to 128. The return_sequences parameter was set to true, indicating that the LSTM layer returns the full length of the sequence at each time step, not just the last output of the last time point. This is important because this data are passed to additional LSTM layers. Finally, we specified a dropout of 0.4. A dropout is a technique that helps avoid overfitting [63] the model.

Then, a BatchNormalization layer [64] is deployed. Its purpose is to stabilize and accelerate neural network learning, especially for deep networks. A BatchNormalization normalizes the inputs of each layer of the neural network, it transforms them in such a way

that the mean becomes 0 and the variance becomes 1. This helps to distinguish between data scaling differences and also stabilizes the distribution of the data. A BatchNormalization layer allows the normalization of current outputs not only for an entire dataset (all examples), but within a minibatch (small set of examples). This means that it uses a separate mean and standard deviation for each minibatch [65,66]. This contributes to the stability of the model calculations. After normalizing the actual input data, it applies weights and offsets to the original data so that the network can learn optimal transformations. By using small minibatches, this layer makes the network somewhat stochastic, which can facilitate regularization and avoid overfitting. We then repeated the LSTM and BatchNormalization layers twice with the same parameters [67].

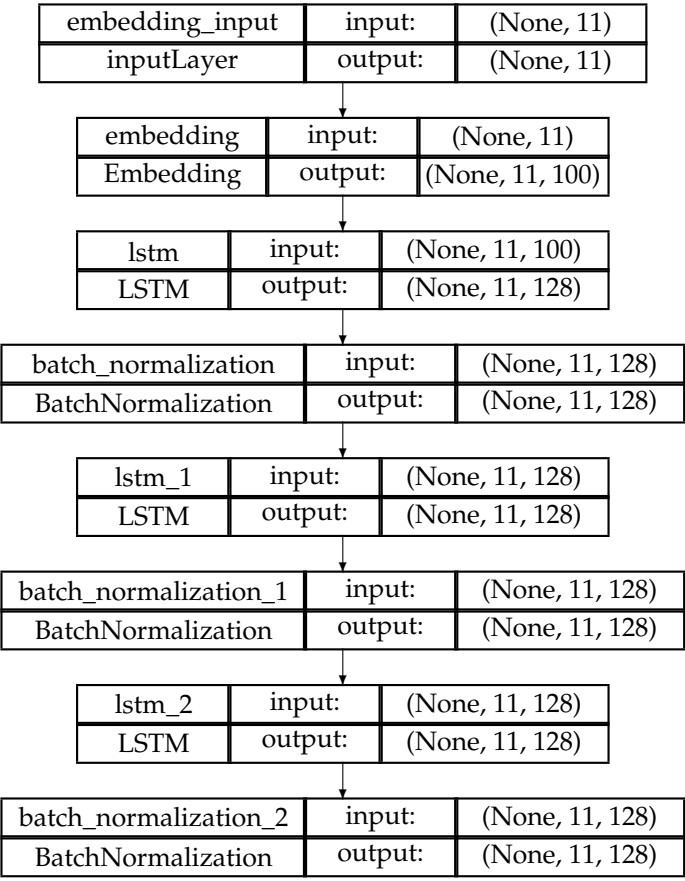

**Figure 2.** The structure of the LSTM model, part 1.

A GlobalAveragePooling1D layer [68,69] is applied after the last LSTM layer, as shown in Figure 3. GlobalAveragePooling1D is designed to convert the output of 2D layers into a simple vector. The "1D" indicates that this method is one-dimensional, i.e., a time series or text dimensions. This layer averages the output time series and returns a single vector containing the averaged values. By averaging the long time series, the GlobalAveragePooling1D layer can produce a small dimension representation of the input texts. This makes the network input length independent [70]. Then, in the next step, a Dense layer [71] with 64 neurons is added and a ReLU [72,73] activation function. This is followed by a Dropout layer [74] with a rate of 0.4. These two layers were repeated a second time in the model structure. The last layer is a SoftMax activation [75,76] layer with 36 neurons, because this is a classification task with 36 different diseases.

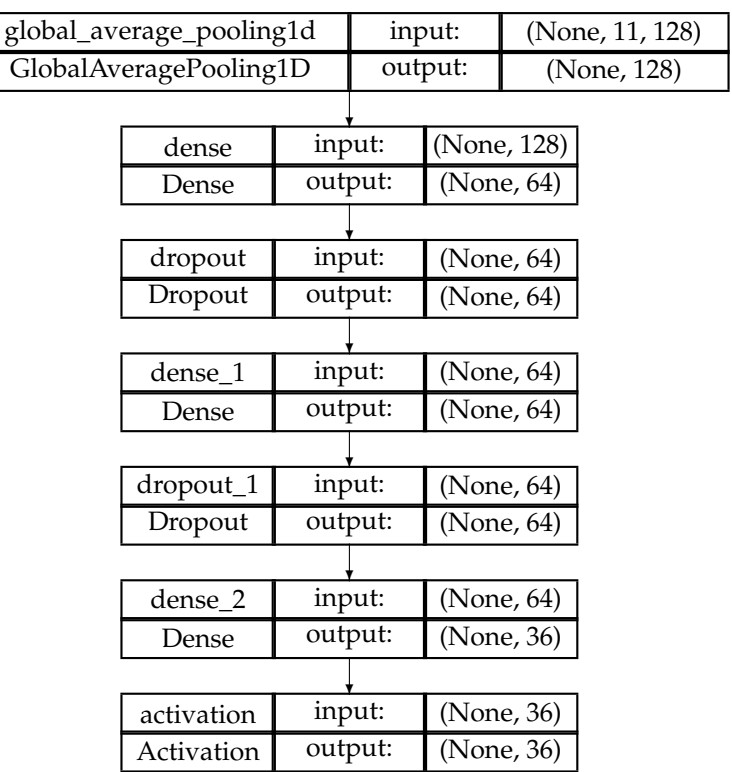

**Figure 3.** The structure of the LSTM model, part 2.

5.1.3. Optimization and Loss Function

Adam optimization [77] was applied to the LSTM model. Adam is one of the optimization algorithms in machine learning, which is mainly used for neural networks. Adam is an acronym that stands for Adaptive Moment Estimation, and derives its name from the fact that the algorithm adapts the learning rate to each weight separately. Adam initializes the weights and a moving average for each of the weights and the square of the weights. These values are set to zero or other initial values. The algorithm uses a minimum size (minibatch) of the training data and then calculates the error and Adam updates the moving averages of the weights and the gradients. This allows the learning rate to vary for each weight separately, which can result in more efficient learning. The algorithm updates the weights according to the learning rate and gradient and then returns to minibatch processing again [78]. For this model, categorical cross-entropy was used [79]. This is a cost function in machine learning that most often is used in classification tasks where classes are categorical or discrete. The input model generates probabilities for all possible classes based on the input data. These probabilities are obtained as the output of the SoftMax [80] activation layer and add up to 1 for all inputs. The real class labels encode which class is the correct one, and categorical cross-entropy compares the estimated probabilities with the real class labels. Categorical cross-entropy is a scalar function that reflects how much the probability distribution of the model differs from the real label distribution. The smaller the value, the better the model fits the real classes. Gradient descent algorithms (such as Adam or Stochastic Gradient Descent) tend to minimize the categorical cross-entropy function during the training of the model [81].

*5.2. Bidirectional Encoder Representations from Transformers*

Bidirectional Encoder Representations from Transformers (BERT) [82–84] stands out as a widely utilized language model crafted by Google Research. Being grounded in the transformer architecture, BERT diverges from sequential text processing, opting for an

attention mechanism to discern word relationships. This parallel processing capability enables transformers to adeptly manage long-term dependencies and comprehend context in a more holistic manner. A pivotal aspect of BERT lies in its bidirectional training methodology. Unlike previous models that adopted a unidirectional approach, relying solely on preceding words to predict the next, BERT employs a masked language model objective in its pre-training. In this process, certain words in the input sentence are randomly masked, and the model is crafted by predicting these masked words within their contextual framework. BERT represents a significant advancement in pre-trained language models, facilitating fine-tuning for diverse tasks such as text classification and question answering. During fine-tuning, the BERT model engages with a labeled dataset specific to the target task, incorporating a task-specific output layer. Leveraging the knowledge acquired from prior tasks, BERT exhibits remarkable performance. Since its inception, various iterations and enhancements have emerged in the realm of language models.

BERT is a transformer model that is pre-trained on a large text source, after which we can easily apply it to our own task. Its goal is to build deep, bidirectional representations of unlabeled text in advance, taking into account left and right context together in each layer. In this way, the pre-trained BERT model can be fine-tuned with a single additional output layer, so that it can be created for a variety of tasks, such as question answering and language inference, without much need for modification. BERT outperforms previous models in natural language processing tasks. For this reason, it has become very widely used in the world of artificial intelligence, as well as in academia. These are the reasons why we chose BERT for our research. The BERT model is capable of syntactic and semantic analysis of human language, and the results it produces are among the best available. The BERT model can take into account all the words in a text and link them to other words in the text. The BERT model is very versatile and can be used for many different tasks. It has achieved many results, one of many being its performance in the SWAG competition. BERT has outperformed previous top models, including human-level performance [57].

Choosing a BERT-based model over GPT for our medical assistant chatbot can be justified for several reasons. BERT is designed for various NLP tasks, including classification, making it well-suited for our specific use case. Unlike GPT [85–88], which is primarily focused on generating coherent and contextually relevant text, BERT's bidirectional architecture allows it to capture intricate relationships between words and better understand the context of medical queries. BERT's pre-training on large corpora helps it grasp nuanced language patterns, aiding performance in classification tasks with smaller datasets. GPT, on the other hand, may not be as effective in scenarios with limited labeled examples. Additionally, BERT's attention mechanism allows it to focus on relevant parts of the input sequence, which is crucial for understanding medical terminology and context. BERT's fine-tuning capabilities make it adaptable to specific domains, allowing our chatbot to learn from the limited data available for medical assistance. GPT's generative nature might not be as well-suited for fine-tuning on specific tasks with a small dataset.

### 5.2.1. Composition

We started preparing the data for the model by taking all input complaints and requests for a specific disease, so that, later, the AI would know what to do to remedy the problem. The next step was to specify the diseases for which it would stick to the correct solutions. We also had to adapt the data to the BERT model so that we could use it correctly. We converted the categorical labels into numerical representations. Each category of input data was assigned a unique integer. We tokenized them and loaded them into the BERT model, which allowed us to use them for various natural language processing tasks that we needed for the chatbot. We used a built-in model, changing the dropout of the hidden layer from the default 0.1 to 0.2. For the model, we needed the length distribution of the input data to determine the maximum length of a sentence that a user could type. Using the TensorDataset, we combined the input sequences, attention [89–91], masks, and labels, and then created a DataLoader with a given batch size and a random sampler to create a

data channel. This pipeline iterated over the training data batch by batch, introducing data into the model for training and optimization.

5.2.2. Layers

A modified BERT model, exhibited in Figure 4, has been devised specifically for enhancing the accuracy of the medical assistant chatbot. This class incorporates BERT as its foundational model, complemented by additional layers to optimize performance. BERT introduces the notion of contextual word representations, signifying the capture of meaning and context for each word based on its surrounding words. The self-monitoring mechanism inherent in BERT's transformer enables it to selectively focus on various segments within the input sequence, emphasizing the relationships between words. Through the utilization of an attention mechanism, BERT adeptly models intricate linguistic structures and dependencies. The model operates by taking input sentence identifiers and attention masks, which are then passed through the BERT model, culminating in the generation of the output [92].

We then added a linear layer, which defines linear transformations of the given input and output dimensions, with 1024 and 768 parameters. The numbers 1024 and 768 denote the input and output dimensions of the fully connected layer. In this case, 1024 corresponds to the input size, which is equal to the dimension of the BERT embeddings. BERT models typically output contextualized word embeddings of 1024 or 768, which capture the meaning and context of each word in the input sentence. The 768 represents the output size, which is the desired dimensionality of the output tensor produced by the fully connected layer. This value can be chosen based on the specific requirements of a given task or as a design decision of the neural network architecture. The fully connected layer takes an input tensor of size [batch_size, 1024] and produces an output tensor of size [batch_size, 768] by performing a linear transformation and applying weights and biases to the input.

Following this, the implementation of the Rectified Linear Unit (ReLU) activation function takes place. This activation function is characterized by numerous advantages, making it widely popular. Its incorporation introduces nonlinearity into the neural network, enhancing the model's ability to recognize and portray intricate relationships. Compared to alternative activation functions like sigmoid [93] or tanh [94], ReLU is a straightforward choice. The ReLU function, defined as $\text{ReLU}(x) = \max(0, x)$, exclusively retains positive values and assigns negative values to zero. This straightforwardness enhances computational efficiency, promoting faster convergence during the training process [95].

The utilization of the ReLU activation function served the purpose of mitigating the risk of vanishing gradients. A vanishing gradient refers to the inefficient transfer of gradients from the model's output back to layers situated closer to the input end in a multilayer neural network [96]. Multilayer models are susceptible to drawing incorrect conclusions due to this phenomenon. ReLU addresses this issue by setting the gradient to zero when it experiences an exponential decrease. This action maintains a constant gradient for positive values, preventing rapid gradient decline and facilitating a smoother gradient flow during the backpropagation process [97]. When the gradient is set to zero, it is disregarded by the model during the training phase. ReLU, by preserving the gradient for positive values, contributes to enhanced generalization performance across various deep learning tasks. By introducing nonlinearity [98] to the model, it allows for more effective results on previously unseen data, marking one of its key attributes.

Subsequently, a Dropout layer was introduced to the model with a dropout rate set at 0.2. Dropout serves as a regularization technique employed in neural networks to mitigate overfitting. This method randomly omits neurons in each training cycle, a percentage specified by the dropout rate. Overfitting occurs when a model excels on training data but struggles to generalize to unseen data. Dropout addresses this challenge by randomly excluding neurons, preventing them from co-adapting excessively. This, in turn, compels individual neurons to enhance their information content and diminish their reliance on

the presence of other neurons. Following this, the final three layers were duplicated.The adjustment was made to accommodate the specific requirements of 36 disease classes.

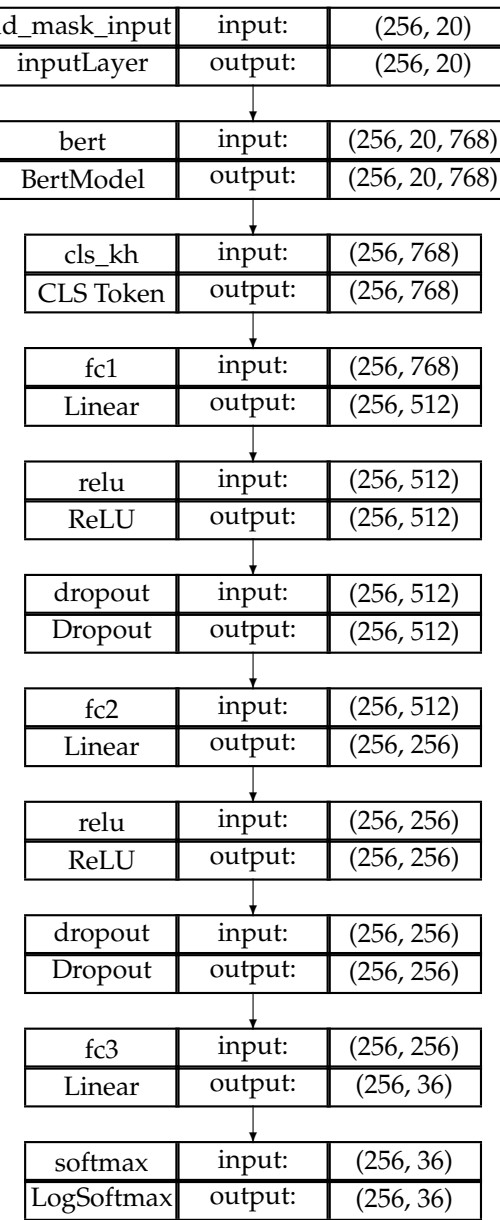

**Figure 4.** The structure of the BERT model.

The final layer incorporates a LogSoftmax activation function. A LogSoftmax is a component commonly used in neural network architectures, particularly in the context of deep learning and machine learning. It is often employed as the final layer in a network for multiclass classification tasks. The LogSoftmax applies the logarithm of the softmax function to the raw output scores (logits) produced by the preceding layers of a neural network, according to the following formula:

$$\text{Logsoftmax}(x)_i = \log\left(\frac{e^{x_i}}{\sum_j e^{x_j}}\right).$$

The negative values produced by the LogSoftmax are not used directly; rather, they are used in the computation of the loss during training. The negative log-likelihood loss, when

combined with the LogSoftmax, provides a measure of how well the predicted probabilities match the true distribution of the classes.

The SoftMax function is a mathematical operation that takes a vector of real numbers and transforms it into a probability distribution, where each element in the vector represents the likelihood of a corresponding class [99].

### 5.2.3. Optimization and Loss Function

The optimization strategy employed for the model was AdamW, a variant of the Adam optimizer commonly applied in conjunction with transformer-based models like BERT. The selection of AdamW was specifically geared towards its compatibility with such models. The optimizer's learning rate was explicitly set to $10^{-3}$, dictating the pace at which the optimizer adjusts the model parameters during training. For handling class imbalance in the classification task, class weights were determined using the sklearn.utils.class_weight module. The "balanced"option was utilized, automatically computing weights inversely proportional to the class frequencies in the input data. This ensures that less frequent classes receive higher weights, addressing the issue of class imbalance.

The negative log-likelihood loss (NLLLoss) [100] function was applied. It is a loss function used in machine learning, particularly in the context of classification problems where the goal is to predict a class label for a given input. This loss function is often used in conjunction with the LogSoftmax activation function in the output layer of a neural network. The NLLLoss is designed to be used with models that output log probabilities, typically obtained by applying the LogSoftmax activation to the raw output scores (logits) [101] of a neural network. The intuition behind the negative log likelihood loss is to penalize models more when they assign low probability to the target class [67].

### 6. Results

Figure 5 demonstrates the validation and training accuracy of our LSTM model, showcasing a significant achievement in model performance. As is exhibited, we have achieved very promising results with the LSTM model. We got 0.91 validation accuracy. We also tested the global F1-score, precision and recall, which came out to 0.9. This shows a very good performance. Since the F1-score examines the correlation between precision and recall, it can be seen that the model performance is very balanced for our data.

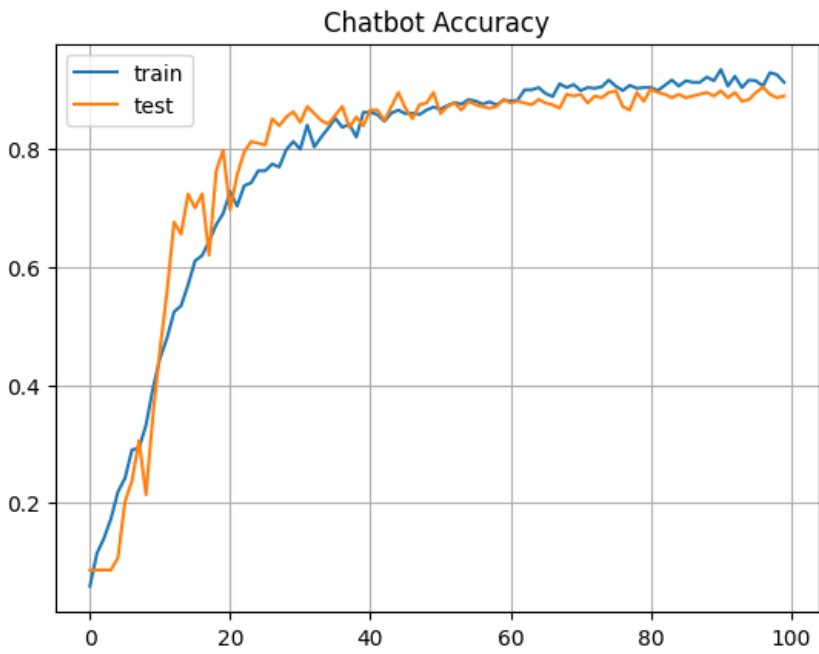

**Figure 5.** The accuracy and validation accuracy of the LSTM model.

Figure 6 shows the training and validation accuracy of a chatbot over 100 epochs. The blue line represents the training accuracy, which stabilizes around 0.75 and slightly improves to 0.8. The orange line for validation accuracy also stabilizes around 0.65 and modestly increases to about 0.7. As it is shown, BERT has very good results, but it is still in its start-up phase. If a sufficient number of complaints are entered for a particular disease, it detects it very well. However, those with even less data are mistaken and not recognized. We also ran into the interesting fact that the complaint given can be the symptom of a wide range of diseases, so it cannot categorize it exactly in the same class as the one we gave it, but it does make the correct deduction. For the near future, we would definitely like to expand the database in two aspects. One is that we want to collect more input data for those diseases that are difficult to recognize. We also want to continue to include other diseases, especially those that are very common in everyday life. As we mentioned with the BERT model, we have so far achieved a validation accuracy of 0.7. As the LSTM model produced significantly better results, we carefully reviewed its results and we would like to present them.

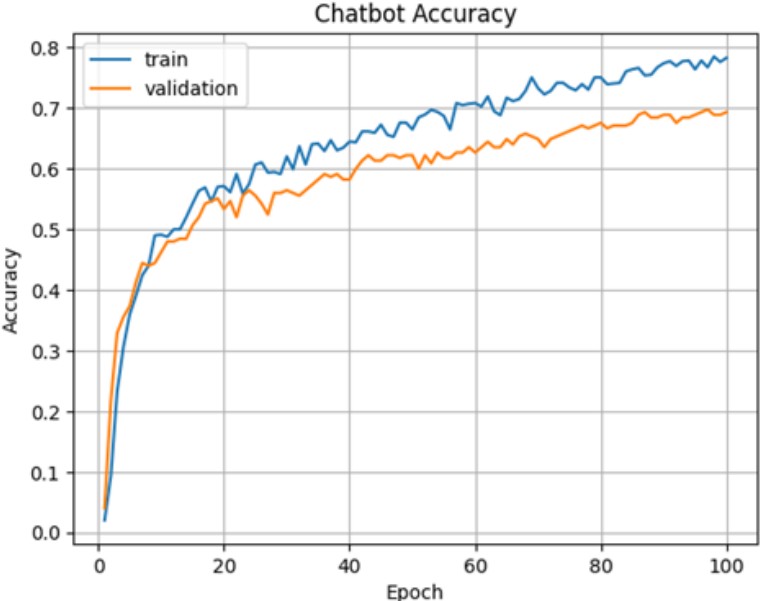

**Figure 6.** The accuracy and validation accuracy of the BERT model.

The F1-score [102] serves as a crucial statistical metric commonly employed in multiclass classification scenarios, particularly in situations where class distribution is imbalanced. This metric is calculated as the harmonic mean of precision and recall, making it a valuable measure for assessing overall classification performance across multiple classes. The F1-score can be calculated using the following formula:

$$\text{F1-score} = 2 \cdot \frac{\text{precision} \cdot \text{recall}}{\text{precision} + \text{recall}},$$

where precision is the number of true positives (TP) [103] divided by the total number of cases classified as positive. This indicates how accurate the classification is for positive results. Recall is the number of TP cases divided by the total number of true positives (true positive + false negative). The advantage of the F1-score is that it is an indicator that considers both indicators in the same way. As precision and recall are often in conflict (high precision for low sensitivity and vice versa), the F1-score helps to find a balanced performance in the classification system. The goal is to achieve a high F1-score, which means that the system classifies cases accurately and efficiently [104].

The following are the diseases, in order: low blood pressure, angioedema, arthritis, chicken pox, fungal skin, COVID-19, vitamin D deficiency, diabetes, eczema, sprains, sore

tooth, earache, ringing in the ears, weakness, bite, bruise, bronchial asthma, dehydration, tearing, conjunctivitis, sunburn, fever, high blood pressure, cold, menstruation, migraine, nasal congestion, nasal flushing, reflux, heart attack, sore throat, pneumonia, cut, anemia, bleeding, and insomnia.

As it is shown in Table 3, the model exhibits varying levels of performance across different classes, demonstrating superior accuracy in certain instances, while showing deficiencies in others. Incomplete data pose a significant challenge, leading to a lack of comprehensive understanding in some cases. As the research progresses, it becomes imperative to incorporate a representative test set from each class during the later stages. Nonetheless, it is important to note that the inclusion of additional data is expected to mitigate the risk associated with these limitations.

**Table 3.** Metrics of the LSTM model's classes.

| Class | 1 | 2 | 3 | 4 | 5 | 6 | 7 | 8 | 9 | 10 | 11 | 12 |
|---|---|---|---|---|---|---|---|---|---|---|---|---|
| Precision | 1.0000 | 1.0000 | 1.0000 | 0.6667 | 0.7500 | 1.0000 | 1.0000 | 0.7778 | 1.0000 | 0.0000 | 0.8000 | 1.0000 |
| Recall | 0.9412 | 1.0000 | 1.0000 | 0.5000 | 1.0000 | 0.9655 | 1.0000 | 1.0000 | 0.6364 | - | 1.0000 | 1.0000 |
| F1-score | 0.9697 | 1.0000 | 1.0000 | 0.5714 | 0.8571 | 0.9824 | 1.0000 | 0.8750 | 0.7778 | - | 0.8889 | 1.0000 |
| Class | 13 | 14 | 15 | 16 | 17 | 18 | 19 | 20 | 21 | 22 | 23 | 24 |
| Precision | 1.0000 | 1.0000 | - | 0.0000 | 0.6250 | 1.0000 | 1.0000 | 1.0000 | 0.0000 | 1.0000 | 1.0000 | 0.0000 |
| Recall | 0.9333 | 0.9091 | 0.0000 | - | 0.8333 | 1.0000 | 1.0000 | 1.0000 | - | 0.7778 | 0.9286 | 0.0000 |
| F1-score | 0.9655 | 0.9524 | - | - | 0.7143 | 1.0000 | 1.0000 | 1.0000 | - | 0.8750 | 0.9630 | 0.0000 |
| Class | 25 | 26 | 27 | 28 | 29 | 30 | 31 | 32 | 33 | 34 | 35 | 36 |
| Precision | 1.0000 | 1.0000 | 1.0000 | 0.0000 | 1.0000 | 1.0000 | 0.0000 | 0.3333 | 0.0000 | 1.0000 | 0.0000 | 0.4000 |
| Recall | 1.0000 | 1.0000 | 0.2500 | - | 1.0000 | 1.0000 | - | 1.0000 | - | 1.0000 | - | 1.0000 |
| F1-score | 1.0000 | 1.0000 | 0.4000 | - | 1.0000 | 1.0000 | - | 0.5000 | - | 1.0000 | - | 0.5714 |

We also examined the Cohen's kappa value ($\kappa$) [105], which is a statistical indicator that is often used to determine the degree of agreement or similarity, especially cases where categorical or discrete variables are evaluated or classified. Establishing the degree of agreement between classifiers can help determine how stable classifications are. If $\kappa$ is high, it indicates that the classifiers have a higher degree of agreement on the classifications. The value of $\kappa$ helps us understand how much the classifications deviate from chance. Cohen's Kappa can range from $-1$ to 1 and indicates the extent to which observers agree on the classification, taking into account chance matches including covariates. It can be calculated as follows:

$$\kappa = \frac{P_o - P_e}{1 - P_e},$$

where $P_o$ represents the observed agreement between observers, while $P_e$ stands for the expected agreement by chance. In the ideal case, where observers are in perfect agreement (no difference between their classifications), $\kappa$ will be 1. When observers are classified completely at random, $\kappa$ is 0. Furthermore, if the observers are worse matched, than would be expected by chance, then $\kappa$ is negative. We obtained a value of 0.9 with the LSTM model, which means that the observers were fairly consistent in their classification [106].

A confusion matrix [107] is a commonly employed matrix in classification tasks, serving as a tool to evaluate the effectiveness of an algorithm or model in carrying out classification assignments.

The analysis of the confusion matrix reveals notable patterns and challenges within the classification model. The main diagonal of the confusion matrix, as can be seen in Table A1, corresponds to instances where the model correctly classified health conditions.

Notably, in departments with a substantial volume of data, the model demonstrates a high level of accuracy, indicating its proficiency in predicting certain diseases.

We have organized a selection of diseases into smaller clusters to assess how they are distinguished by the model, focusing on respiratory issues.

These diseases in Table 4 either share similar symptoms or one condition may be a subset of another. It is evident that the model struggles to accurately define "colds", often confusing it with nasal congestion and fever. Interestingly, while it does not consistently identify a sore throat, it also does not misclassify it with these conditions. Nasal congestion is frequently mistaken for fever, yet the model reliably identifies fever correctly. Accurate differentiation is important for diagnosis and treatment, as different respiratory diseases may require different treatment. For example, the choice of the right therapy depends on whether someone is suffering from a cold, pneumonia, or another respiratory problem.

The next category we looked at was skin problems.

**Table 4.** Confusion matrix of respiratory issues.

| | | | | | | |
|---|---|---|---|---|---|---|
| Colds | 0 | 0 | 0 | 0 | 5 | 2 |
| Pneumonia | 0 | 1 | 0 | 0 | 0 | 0 |
| Sore throat | 0 | 0 | 0 | 0 | 0 | 0 |
| Nasal flushing | 0 | 0 | 0 | 0 | 1 | 0 |
| Nasal congestion | 0 | 0 | 0 | 0 | 2 | 3 |
| Fever | 0 | 0 | 0 | 0 | 0 | 7 |
| | Colds | Pneumonia | Sore throat | Nasal flushing | Nasal congestion | Fever |

It is noticeable in Table 5 that chickenpox is distinctly recognized and not confused with other conditions. However, there are instances where the model incorrectly identifies skin fungus as chickenpox. Conversely, eczema is consistently classified accurately. Despite this, the model consistently misidentifies sunburn but does not conflate it with other skin diseases in this category. Differentiation of skin diseases is essential for a therapeutic approach and prognosis, as different etiologies and clinical features of diseases require different treatment protocols. An accurate diagnosis helps clinicians to apply targeted therapeutic strategies, thereby minimizing the potential complications and exacerbations of untreated or inappropriately treated skin diseases. We have dealt with accident-related problems in the following groupings.

**Table 5.** Confusion matrix of skin problems.

| | | | | |
|---|---|---|---|---|
| Chickenpox | 2 | 0 | 0 | 0 |
| Fungal skin | 2 | 6 | 0 | 0 |
| Eczema | 0 | 0 | 7 | 0 |
| Sunburn | 0 | 0 | 0 | 0 |
| | Chickenpox | Fungal skin | Eczema | Sunburn |

Here. it can be seen in Table 6 that the model occasionally confuses a sprain with a bite, consistently misclassifying the latter. However, it does not erroneously classify bites alongside other complaints. Notably, it consistently predicts bites within other classes. This confusing result may suggest that the model is not able to clearly or effectively separate and classify individual diagnoses. This may be because the similarities or differences between diagnoses are not clear or consistent, or the model may not have sufficient data or ability to accurately distinguish between them. It is also possible that the model does not have sufficient information about the specificities or characteristics of the diagnoses, which

can lead to confusing results. In our case, the lack of data is probably the main problem. For these accident problems, the advice given by the medical chatbot is very important. For example, in the case of a bite or bite wound, it may be important to identify the type of animal and administer antibiotics immediately to prevent wound infection. However, a torn muscle or tendon may require rest and physiotherapy to heal.

Diseases of the circulatory system are part and parcel of our everyday lives, so we have paid close attention to them.

**Table 6.** Confusion matrix of accident-related problems.

| | | | | | |
|---|---|---|---|---|---|
| Sprain | 0 | 1 | 0 | 0 | 0 |
| Bite | 0 | 0 | 0 | 0 | 0 |
| Bruise | 0 | 1 | 0 | 0 | 0 |
| Cut | 0 | 1 | 0 | 0 | 0 |
| Bleeding | 0 | 2 | 0 | 0 | 0 |
| | Sprain | Bite | Bruise | Cut | Bleeding |

In this instance, in Table 7, it is evident that there is no confusion between these diseases, as indicated by a perfect sub-matrix. Additionally, neither of these classes is misclassified with any other diseases, with the model consistently making accurate predictions. Correctly distinguishing them is key to choosing the appropriate medical intervention and treatment, as these conditions require different therapeutic strategies. In the case of low or high blood pressure, timely treatment can significantly improve vitality and quality of life, while in the case of heart attack or anemia, immediate intervention is necessary to avoid serious complications.

**Table 7.** Confusion matrix of diseases of the circulatory system.

| | | | | |
|---|---|---|---|---|
| Hypotension | 16 | 0 | 0 | 0 |
| Hypertension | 0 | 13 | 0 | 0 |
| Heart attack | 0 | 0 | 13 | 0 |
| Anemia | 0 | 0 | 0 | 8 |
| | Hypotension | Hypertension | Heart attack | Anemia |

In Appendix A, the full 36 × 36 confusion matrix can be found, where some other minor mistakes can be observed. There, for example, a lot of classes were misclassified as bites, including those mentioned in Table 6. Alternatively, there were instances where the model incorrectly classified a sore tooth and pneumonia as eczema.

The metrics used in the development of the medical assistant chatbot, such as accuracy, F1-score, confusion matrix, and Cohen's Kappa, play a key role in evaluating the performance of the model. These metrics are not only general statistical indicators, but also provide a deeper understanding of the system's application and effectiveness in a medical environment. In this case, it is critical that the model accurately interprets and classifies medical terms or symptoms. High accuracy means that the chatbot reliably recognizes the input, which is essential for medical advice and information transfer. F1-score in this area means that rare or less common symptoms are accurately recognized by the chatbot, which increases the performance of the system. In the case of the medical assistant, it is essential to observe which symptoms or diagnoses are easily confused by the system and to pay particular attention to these when further refining the system, and, therefore it is worth using a confusion matrix. For medical texts, where an input may belong to more than one category (e.g., several symptoms at the same time), Cohen's Kappa helps to evaluate the consistency of the system. Using these metrics together helps ensure that the chatbot not only performs well in general, but is also effective when tailored specifically to the specifics

of the medical field. These evaluations help to identify weaknesses in the model and allow for further refinements and improvements to improve the system's medical applicability.

## 7. Conclusions

In addition, our analysis revealed that the inherent intricacies of BERT's attention mechanisms, which, while advantageous in capturing contextual nuances in large corpora, may have posed challenges in effectively adapting to the limited scope of our dataset. Furthermore, the fine-tuning process of BERT demands a substantial amount of annotated data to harness its full potential, which was lacking in our current experimental setup. Nonetheless, despite these limitations, the discernible efficacy of BERT in our preliminary findings underscores its potential utility as a cornerstone for future iterations of our project. As we accrue more diverse data samples and refine our model architecture, we anticipate that the latent capabilities of BERT will manifest more prominently, ultimately yielding superior performance in our target NLP tasks.

## 8. Discussion

While our study has advanced the understanding of AI tool development for the Hungarian language, it is important to acknowledge its limitations and suggest avenues for future research.

One of the principal limitations of this study is its reliance on available datasets, which are not as comprehensive or diverse as those for more widely studied languages. This scarcity of resources could affect the generalizability of our findings and may limit the effectiveness of the proposed AI models. Looking ahead, future research should focus on expanding the quantity and quality of linguistic resources for Hungarian. This includes the creation of larger, more diverse corpora that are richly annotated with morphological, syntactic, and semantic information.

Furthermore, exploring the application of newer AI techniques, such as deep learning architectures that have shown promise in other agglutinative languages, could provide breakthroughs in the processing of Hungarian [108]. Implementing and testing these technologies could help overcome some of the morphological and syntactic processing challenges identified in this study.

By addressing these limitations and following the suggested future directions, we can enhance the efficacy and reach of AI technologies, ensuring that they serve the needs of the Hungarian-speaking community more effectively. This approach not only aids in language preservation but also enriches the linguistic diversity and technological robustness of AI applications globally.

**Author Contributions:** All Authors equally contributed to the work. All authors have read and agreed to the published version of the manuscript.

**Funding:** Project no. 2019-1.3.1-KK-2019-00007 was implemented with the support provided from the National Research, Development and Innovation Fund of Hungary, financed under the 2019-1.3.1-KK funding scheme. This project was supported by the National Research, Development, and Innovation Fund of Hungary, financed under the TKP2021-NKTA-36 funding scheme. The work of L. Szilágyi was partially supported by the Consolidator Researcher Program of Óbuda University.

**Institutional Review Board Statement:** Not applicable.

**Informed Consent Statement:** Not applicable.

**Data Availability Statement:** Access to the data is available upon request. Access to the data can be requested via e-mail to the corresponding author.

**Acknowledgments:** On behalf of the AI development for diabetic and brain MRI scans project, we are grateful for the possibility to use ELKH Cloud (see Héder et al. 2022; Available online: https://science-cloud.hu/ accessed on 1 January 2022 ), which helped us achieve the results published in this paper.

**Conflicts of Interest:** The authors declare no conflicts of interest.

## Abbreviations

The following abbreviations are used in this manuscript:

| | |
|---|---|
| NLP | Natural language processing |
| MTS | Manchester Triage System |
| GDPR | General Data Protection Regulation |
| SVM | Support Vector Machine |
| LSTM | Long Short-Term Memory |
| RNN | Recurrent Neural Network |
| BERT | Bidirectional Encoder Representations from Transformers |
| ADAM | Adaptive Moment Estimation |
| GPT | Generative Pre-trained Transformers |
| AI | Artificial intelligence |
| ReLU | Rectified Linear Unit |
| NLLLoss | Negative log-likelihood loss |
| TP | True positive |
| AUC | Area Under the ROC Curve |
| KNN | K-Nearest Neighbors |

## Appendix A. Confusion Matrix of the LSTM Model

**Table A1.** The entire confusion matrix of the LSTM model.

| | | | | | | | | | | | | | | | | | | | | | | | | | | | | | | | | | | | |
|--|--|--|--|--|--|--|--|--|--|--|--|--|--|--|--|--|--|--|--|--|--|--|--|--|--|--|--|--|--|--|--|--|--|--|--|
| 16 | 0 | 0 | 0 | 0 | 0 | 0 | 0 | 0 | 0 | 0 | 0 | 0 | 0 | 0 | 0 | 0 | 0 | 0 | 0 | 0 | 0 | 0 | 0 | 0 | 0 | 0 | 0 | 0 | 0 | 0 | 0 | 0 | 0 | 0 | 0 |
| 0 | 7 | 0 | 0 | 0 | 0 | 0 | 0 | 0 | 0 | 0 | 0 | 0 | 0 | 0 | 0 | 0 | 0 | 0 | 0 | 0 | 0 | 0 | 0 | 0 | 0 | 0 | 0 | 0 | 0 | 0 | 0 | 0 | 0 | 0 | 0 |
| 0 | 0 | 22 | 0 | 0 | 0 | 0 | 0 | 0 | 0 | 0 | 0 | 0 | 0 | 0 | 0 | 0 | 0 | 0 | 0 | 0 | 0 | 0 | 0 | 0 | 0 | 0 | 0 | 0 | 0 | 0 | 0 | 0 | 0 | 0 | 0 |
| 0 | 0 | 0 | 2 | 0 | 0 | 0 | 0 | 0 | 0 | 0 | 0 | 0 | 0 | 1 | 0 | 0 | 0 | 0 | 0 | 0 | 0 | 0 | 0 | 0 | 0 | 0 | 0 | 0 | 0 | 0 | 0 | 0 | 0 | 0 | 0 |
| 0 | 0 | 0 | 2 | 6 | 0 | 0 | 0 | 0 | 0 | 0 | 0 | 0 | 0 | 0 | 0 | 0 | 0 | 0 | 0 | 0 | 0 | 0 | 0 | 0 | 0 | 0 | 0 | 0 | 0 | 0 | 0 | 0 | 0 | 0 | 0 |
| 0 | 0 | 0 | 0 | 0 | 28 | 0 | 0 | 0 | 0 | 0 | 0 | 0 | 0 | 0 | 0 | 0 | 0 | 0 | 0 | 0 | 0 | 0 | 0 | 0 | 0 | 0 | 0 | 0 | 0 | 0 | 0 | 0 | 0 | 0 | 0 |
| 0 | 0 | 0 | 0 | 0 | 0 | 12 | 0 | 0 | 0 | 0 | 0 | 0 | 0 | 0 | 0 | 0 | 0 | 0 | 0 | 0 | 0 | 0 | 0 | 0 | 0 | 0 | 0 | 0 | 0 | 0 | 0 | 0 | 0 | 0 | 0 |
| 1 | 0 | 0 | 0 | 0 | 0 | 0 | 7 | 0 | 0 | 0 | 0 | 0 | 0 | 1 | 0 | 0 | 0 | 0 | 0 | 0 | 0 | 0 | 0 | 0 | 0 | 0 | 0 | 0 | 0 | 0 | 0 | 0 | 0 | 0 | 0 |
| 0 | 0 | 0 | 0 | 0 | 0 | 0 | 0 | 7 | 0 | 0 | 0 | 0 | 0 | 0 | 0 | 0 | 0 | 0 | 0 | 0 | 0 | 0 | 0 | 0 | 0 | 0 | 0 | 0 | 0 | 0 | 0 | 0 | 0 | 0 | 0 |
| 0 | 0 | 0 | 0 | 0 | 0 | 0 | 0 | 0 | 0 | 0 | 0 | 0 | 0 | 1 | 0 | 0 | 0 | 0 | 0 | 0 | 0 | 0 | 0 | 0 | 0 | 0 | 0 | 0 | 0 | 0 | 0 | 0 | 0 | 0 | 0 |
| 0 | 0 | 0 | 0 | 0 | 0 | 0 | 0 | 2 | 0 | 8 | 0 | 0 | 0 | 0 | 0 | 0 | 0 | 0 | 0 | 0 | 0 | 0 | 0 | 0 | 0 | 0 | 0 | 0 | 0 | 0 | 0 | 0 | 0 | 0 | 0 |
| 0 | 0 | 0 | 0 | 0 | 0 | 0 | 0 | 0 | 0 | 0 | 14 | 0 | 0 | 0 | 0 | 0 | 0 | 0 | 0 | 0 | 0 | 0 | 0 | 0 | 0 | 0 | 0 | 0 | 0 | 0 | 0 | 0 | 0 | 0 | 0 |
| 0 | 0 | 0 | 0 | 0 | 0 | 0 | 0 | 0 | 0 | 0 | 0 | 14 | 0 | 0 | 0 | 0 | 0 | 0 | 0 | 0 | 0 | 0 | 0 | 0 | 0 | 0 | 0 | 0 | 0 | 0 | 0 | 0 | 0 | 0 | 0 |
| 0 | 0 | 0 | 0 | 0 | 0 | 0 | 0 | 0 | 0 | 0 | 0 | 0 | 20 | 0 | 0 | 0 | 0 | 0 | 0 | 0 | 0 | 0 | 0 | 0 | 0 | 0 | 0 | 0 | 0 | 0 | 0 | 0 | 0 | 0 | 0 |
| 0 | 0 | 0 | 0 | 0 | 0 | 0 | 0 | 0 | 0 | 0 | 0 | 0 | 0 | 0 | 0 | 0 | 0 | 0 | 0 | 0 | 0 | 0 | 0 | 0 | 0 | 0 | 0 | 0 | 0 | 0 | 0 | 0 | 0 | 0 | 0 |
| 0 | 0 | 0 | 0 | 0 | 0 | 0 | 0 | 0 | 0 | 0 | 0 | 0 | 0 | 1 | 0 | 0 | 0 | 0 | 0 | 0 | 0 | 0 | 0 | 0 | 0 | 0 | 0 | 0 | 0 | 0 | 0 | 0 | 0 | 0 | 0 |
| 0 | 0 | 0 | 0 | 0 | 0 | 0 | 0 | 0 | 0 | 0 | 0 | 0 | 0 | 1 | 0 | 5 | 0 | 0 | 0 | 0 | 0 | 0 | 1 | 1 | 0 | 0 | 0 | 0 | 0 | 0 | 0 | 0 | 0 | 0 | 0 |
| 0 | 0 | 0 | 0 | 0 | 0 | 0 | 0 | 0 | 0 | 0 | 0 | 0 | 0 | 0 | 0 | 0 | 5 | 0 | 0 | 0 | 0 | 0 | 0 | 0 | 0 | 0 | 0 | 0 | 0 | 0 | 0 | 0 | 0 | 0 | 0 |
| 0 | 0 | 0 | 0 | 0 | 0 | 0 | 0 | 0 | 0 | 0 | 0 | 0 | 0 | 0 | 0 | 0 | 0 | 3 | 0 | 0 | 0 | 0 | 0 | 0 | 0 | 0 | 0 | 0 | 0 | 0 | 0 | 0 | 0 | 0 | 0 |
| 0 | 0 | 0 | 0 | 0 | 0 | 0 | 0 | 0 | 0 | 0 | 0 | 0 | 0 | 0 | 0 | 0 | 0 | 0 | 11 | 0 | 0 | 0 | 0 | 0 | 0 | 0 | 0 | 0 | 0 | 0 | 0 | 0 | 0 | 0 | 0 |
| 0 | 0 | 0 | 0 | 0 | 0 | 0 | 0 | 0 | 0 | 0 | 0 | 0 | 0 | 1 | 0 | 0 | 0 | 0 | 0 | 0 | 0 | 0 | 0 | 0 | 0 | 0 | 0 | 0 | 0 | 0 | 0 | 0 | 0 | 0 | 0 |
| 0 | 0 | 0 | 0 | 0 | 0 | 0 | 0 | 0 | 0 | 0 | 0 | 0 | 0 | 0 | 0 | 0 | 0 | 0 | 0 | 0 | 7 | 0 | 0 | 0 | 0 | 0 | 0 | 0 | 0 | 0 | 0 | 0 | 0 | 0 | 0 |
| 0 | 0 | 0 | 0 | 0 | 0 | 0 | 0 | 0 | 0 | 0 | 0 | 0 | 0 | 0 | 0 | 0 | 0 | 0 | 0 | 0 | 0 | 13 | 0 | 0 | 0 | 0 | 0 | 0 | 0 | 0 | 0 | 0 | 0 | 0 | 0 |
| 0 | 0 | 0 | 0 | 0 | 0 | 0 | 0 | 0 | 0 | 0 | 0 | 0 | 0 | 0 | 0 | 0 | 0 | 0 | 0 | 0 | 0 | 2 | 0 | 0 | 0 | 5 | 0 | 0 | 0 | 0 | 0 | 0 | 0 | 0 | 0 |
| 0 | 0 | 0 | 0 | 0 | 0 | 0 | 0 | 0 | 0 | 0 | 0 | 0 | 0 | 0 | 0 | 0 | 0 | 0 | 0 | 0 | 0 | 0 | 0 | 29 | 0 | 0 | 0 | 0 | 0 | 0 | 0 | 0 | 0 | 0 | 0 |
| 0 | 0 | 0 | 0 | 0 | 0 | 0 | 0 | 0 | 0 | 0 | 0 | 0 | 0 | 0 | 0 | 0 | 0 | 0 | 0 | 0 | 0 | 0 | 0 | 0 | 33 | 0 | 0 | 0 | 0 | 0 | 0 | 0 | 0 | 0 | 0 |
| 0 | 0 | 0 | 0 | 0 | 0 | 0 | 0 | 0 | 0 | 0 | 0 | 0 | 0 | 0 | 0 | 0 | 0 | 0 | 0 | 0 | 0 | 0 | 3 | 0 | 0 | 2 | 0 | 0 | 0 | 0 | 0 | 0 | 0 | 0 | 0 |
| 0 | 0 | 0 | 0 | 0 | 0 | 0 | 0 | 0 | 0 | 0 | 0 | 0 | 0 | 1 | 0 | 0 | 0 | 0 | 0 | 0 | 0 | 0 | 0 | 0 | 0 | 1 | 0 | 0 | 0 | 0 | 0 | 0 | 0 | 0 | 0 |
| 0 | 0 | 0 | 0 | 0 | 0 | 0 | 0 | 0 | 0 | 0 | 0 | 0 | 0 | 0 | 0 | 0 | 0 | 0 | 0 | 0 | 0 | 0 | 0 | 0 | 0 | 0 | 0 | 7 | 0 | 0 | 0 | 0 | 0 | 0 | 0 |
| 0 | 0 | 0 | 0 | 0 | 0 | 0 | 0 | 0 | 0 | 0 | 0 | 0 | 0 | 0 | 0 | 0 | 0 | 0 | 0 | 0 | 0 | 0 | 0 | 0 | 0 | 0 | 0 | 0 | 13 | 0 | 0 | 0 | 0 | 0 | 0 |
| 0 | 0 | 0 | 0 | 0 | 0 | 0 | 0 | 0 | 0 | 0 | 0 | 0 | 0 | 2 | 0 | 0 | 0 | 0 | 0 | 0 | 0 | 0 | 0 | 0 | 0 | 0 | 0 | 0 | 0 | 0 | 0 | 0 | 0 | 0 | 0 |
| 0 | 0 | 0 | 0 | 0 | 0 | 0 | 0 | 2 | 0 | 0 | 0 | 0 | 0 | 0 | 0 | 0 | 0 | 0 | 0 | 0 | 0 | 0 | 0 | 0 | 0 | 0 | 0 | 0 | 0 | 0 | 1 | 0 | 0 | 0 | 0 |
| 0 | 0 | 0 | 0 | 0 | 0 | 0 | 0 | 0 | 0 | 0 | 0 | 0 | 0 | 1 | 0 | 0 | 0 | 0 | 0 | 0 | 0 | 0 | 0 | 0 | 0 | 0 | 0 | 0 | 0 | 0 | 0 | 0 | 0 | 0 | 0 |
| 0 | 0 | 0 | 0 | 0 | 0 | 0 | 0 | 0 | 0 | 0 | 0 | 0 | 0 | 0 | 0 | 0 | 0 | 0 | 0 | 0 | 0 | 0 | 0 | 0 | 0 | 0 | 0 | 0 | 0 | 0 | 0 | 0 | 8 | 0 | 0 |
| 0 | 0 | 0 | 0 | 0 | 0 | 0 | 0 | 0 | 0 | 0 | 0 | 0 | 0 | 2 | 0 | 0 | 0 | 0 | 0 | 0 | 0 | 0 | 0 | 0 | 0 | 0 | 0 | 0 | 0 | 0 | 0 | 0 | 0 | 0 | 0 |
| 0 | 0 | 0 | 0 | 0 | 1 | 0 | 0 | 0 | 0 | 0 | 0 | 0 | 0 | 1 | 0 | 1 | 0 | 0 | 0 | 0 | 0 | 0 | 0 | 0 | 0 | 0 | 0 | 0 | 0 | 0 | 0 | 0 | 0 | 0 | 2 |

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
