# Peer review of "Advancing Medical Assistance: Developing an Effective Hungarian-Language Medical Chatbot with Artificial Intelligence"

_information, doi:10.3390/info15060297_

Round 1

Reviewer 1 Report

Comments and Suggestions for Authors

Strengths of the Paper:

1.       The paper explores the development of a Hungarian-language medical chatbot using artificial intelligence, which is a relevant topic in the field of healthcare technology.

2.       The authors leverage machine learning techniques, such as LSTM and BERT models, to enhance the chatbot's capabilities, showcasing a sophisticated approach to the development of the system.

Weaknesses of the Paper:

1.       The paper mentions challenges related to the limited scope of the dataset, indicating a potential limitation in the model's performance due to insufficient annotated data .

2.       The discussion on varying levels of performance across different classes suggests that there may be deficiencies in the chatbot's understanding of certain medical conditions, highlighting the need for a more comprehensive test set.

3.       The paper lacks a detailed description of the methodology used in developing the chatbot, which could hinder reproducibility and understanding of the research process.

4.       The paper does not extensively discuss the chatbot's ability to generalize to new, unseen data, which is crucial for real-world applications.

5.       There is a lack of discussion on ethical considerations related to the use of AI in healthcare, such as patient privacy, data security, and potential biases in the chatbot's recommendations.

6.       The scalability of the chatbot system for handling a large volume of user queries and maintaining performance under increased load is not addressed in detail.

7.       The paper does not touch upon the user interface design of the chatbot, which plays a significant role in user experience and acceptance.

8.       In terms of the Hungarian-based NLP and speech, it is better to cite this article http://dx.doi.org/10.3311/WINS2024-004

Comments on the Quality of English Language

The quality of the English is clear and easily understandable.

Author Response

Thank you very much for your review. We have tried to make all changes to be a better manuscript.

Reviewer 2 Report

Comments and Suggestions for Authors

The paper is interesting with a potential for publication.

Nonetheless, before this can happen, authors should provide adequate responses, and relative modifications of the paper, to the following issues.

- First it is not that clear the issue about the difficulty to develop an AI tool based on the Hungarian language. Is there any technical linguistic issue related to this specific language or only a fact due to other technical motivation? please discuss;

- The general architecture of the chatboat is explained rather well, nonetheless there is scarce information about both the more specific algorithms employed and the dataset in use. Are they original (algorithms)? Are they open and available to all (dataset)?

- I have another point regarding the datasets in use. It is well known that we have a lot of problem both for ML and DL if the datasets are not well balanced or semantically sloppy. I would like the authors will discuss this issue preferably with reference to the the following paper (to be cited): AA VV An alternative approach to dimension reduction for pareto distributed data: a case study, J Big Data, Volume 8, Issue 1, December 2021, doi: 10.1186/s40537-021-00428-8

- finally, I have found scarce also the bibliographical Section regarding the field of implications of using AI/chatbots. I would like to see at least one or two more modern papers on AI based chatbots implications, like for example, the following:  AA VV, Chatbots in Education and Research: A Critical Examination of Ethical Implications and Solutions, Sustainability 2023, 15(7),; doi: 10.3390/su15075614

Author Response

(The authors gave the same response as above.)

Reviewer 3 Report

Comments and Suggestions for Authors

The research paper presents a medical chatbot for Hungarian speakers, accessible via a website and mobile app. The chatbot provides health advice, recognizes diseases from user input, and suggests treatments. To further enhance the quality, acountibiity, and scientific sounders, here are my suggestions:

1. There is a need for a comparative analysis of the current chatbot with other medical chatbots as presented in the literature (i.e., references [1–5] and beyond). A comparison table would provide a fair comparison of aspects such as functionality, accuracy, interface, and integration support.

2. Some tables and figures are not properly cited in the text. A brief description should be added where they are cited. For example, Table 1, Figure 1, Figure 5, and Figure 6 are not cited in the text and lack detailed descriptions. These should be added for clarity and completeness.

3. The introduction section needs to be expanded to acknowledge the underlying problem, the proposed solution, and the key contributions of this research. It would be beneficial and reader-friendly if the authors could add a paragraph outlining the organization of the paper.

4. Finally, there is a need to declare and acknowledge the limitations of the study and indicate possible future directions. This will provide transparency and list the way for future research in this area.

Comments on the Quality of English Language

Minor editing of English language required

Author Response

(The authors gave the same response as above.)

Round 2

Reviewer 1 Report

Comments and Suggestions for Authors

The authors have satisfactorily replied to the concerns raised and complemented the manuscript accordingly.

Author Response

Thank you for your reviewing.

Reviewer 2 Report

Comments and Suggestions for Authors

I did not see any relevant changes in response to the issues I have raised. Hence I stay neutral wrt this paper. I am not completely enthusiastic about it, honestly. Be the Editor to take a decision.

Author Response

We have done correction of manuscript

Round 3

Reviewer 2 Report

Comments and Suggestions for Authors

Unfortunately,

I still do not see any correspondence between what the authors have written in response to my criticisms and the correspondent modifications (made to the paper) which remains scarce and vague or even absent.

I provide just an example. The following is the comment the authors have proposed in response to one fo the issues I have raised about unbalanced datasets and relative quality:

"In response to the concern about dataset quality, particularly the issues of balance and semantic clarity, it is crucial to consider how these factors impact machine learning (ML) and deep learning (DL) models. Imbalanced datasets can skew model training, leading to biased outputs and poor generalization to real-world scenarios. Similarly, semantically sloppy datasets, where the data is noisy or inconsistently labeled, can confuse models and degrade their performance.

The reference paper, "An alternative approach to dimension reduction for Pareto distributed data: a case study" by AA VV, published in J Big Data (Volume 8, Issue 1, December 2021), offers insights that could be relevant to addressing these challenges in the context of Hungarian language processing. Although the paper primarily focuses on dimension reduction for Pareto-distributed data, its methodologies, and findings could be adapted to improve the handling of unbalanced and semantically inconsistent datasets in NLP tasks.

Specifically, the authors’ approach to dimension reduction, which prioritizes preserving significant variance in highly skewed distributions, could inform techniques for managing datasets where certain linguistic features or labels are disproportionately represented. By integrating such dimensionality reduction techniques, researchers might better manage and interpret large, complex datasets, leading to more robust AI models for languages like Hungarian that face data scarcity and quality issues"

Nothing of what written above is reported in the paper!

Under these circumstances, I wonder if this review makes any sense or not!

Author Response

Sorry, we didn't put this amendment in, we just responded to it. Thank you very much for the clarification. We have now made the change. 

Reference added to manuscript ([42],line 213-229)
